# Reappraisal of Data of Hydrological Changes Associated with Some Strong Historical Italian Earthquakes

Corrado Castellano [ID], Luigi Cucci *[ID] and Andrea Tertulliani [ID]

Istituto Nazionale di Geofisica e Vulcanologia, 00143 Rome, Italy
* Correspondence: luigi.cucci@ingv.it

**Abstract:** Historical seismology retrieves information about the effects of earthquakes that occurred in the past, mostly regarding the damage, but also on environmental effects. In this paper, we describe the methodology of our research on earthquake-induced hydrological effects, which have been long observed and documented, and are among the most outstanding coseismic phenomena. The method of research follows two distinct paths, depending on whether the investigated event occurred before or after the end of the 18th Century. For the most ancient events, we present examples of historical accounts, local reports, private letters, and diaries, in which the information of interest is often hidden within broader descriptions and mentioned as a minor curiosity. On the contrary, for more recent earthquakes, the research benefits from the growing interest in naturalistic observations that marked the onset of the 19th Century, and is achieved through detailed descriptions, journals, seismic postcards, and through the first systematic collections of instrumental data. Finally, we describe a possible method of classification of the hydrological data and show an analysis of the potential applications and outcomes of this type of research.

**Keywords:** historical seismology; earthquakes effects; hydrological data; Southern Apennines; Central Italy

## 1. Introduction

The main commitment of historical seismology is to expand the knowledge of seismicity as far back as possible. To this aim, investigators working in this branch of seismology retrieve information regarding the effects of earthquakes that occurred in pre-instrumental times (until about the first half of the 20th Century) to provide seismological parameters for the compilation of the seismic long-term earthquakes' catalogs [1–5]. Traditionally, historical seismology has been mainly dedicated to the finding of information regarding the damage of buildings and/or the effects on people [6–8], allowing for the estimation of the intensity of an earthquake with macroseismic methods. However, historical research on ancient seismic events even allows for the retrieval of information different from the damage statistics, such as the geomorphological, geological, and hydrological effects. Starting in the late 18th Century, a renewed interest in the occurrence of natural phenomena encouraged a number of scientists and observers to travel to areas stricken by a severe seismic event to investigate the causes of the earthquake and document its effects.

One of these interesting examples was represented by an ante litteram macroseismic questionnaire drawn by the geologist Giacomo Paci, retrieved from the National Archive of Naples [9]. The document deals with a request, put forward by the Central Authorities, for information and data regarding some earthquakes that had been felt in Calabria (Southern Italy). To gather statistical observations regarding earthquake effects, the questionnaire formulated several questions, including *"sonosi osservate screpolature del suolo ed uscirne mofete?"* (transl. Did you observe ground rupturing and emission of mofette?) and *"vi è stata alterazione nel corso delle acque dei fiumi, e dei ruscelli, ed in quelle dei pozzi?"* (transl. Did you observe any variations in the flow of rivers or streams, and in the level of wells?).

Not surprisingly, the observations of natural effects were thus included in the diagnostic lists of the macroseismic scales formulated during the first half of the 20th Century [10–13], and contribute to the definition of the Environmental Seismic Intensity scale (ESI), which was established at the beginning of the 21st Century and only assesses intensity by the use of such effects [14]. The present legacy of the activity of environmental effects collection are accurate and detailed coeval reports, which are now available to modern investigators (e.g., [15–19], and for only some of the strongest earthquakes that occurred in Italy). The observations retrieved from those reports represent the starting point of studies dedicated to seismotectonic analyses [20–27], to hazard estimates [28,29], or to the compilation of specific catalogs [23,28,30–33]. In particular, hydrological effects created by earthquakes have long been observed and documented, as they are among the most outstanding coseismic phenomena and can be observed over great distances [34,35]. Examples include increases or decreases in streamflow, variations in the water levels in wells, formation and/or disappearance of springs and changes in their discharge, and changes in the chemical/physical characteristics of waters. In the last decades the character of the coseismic hydrological changes has often been found to be related to the style of faulting [35–38]; in this way, the contribution of historical data of earthquake-induced hydrological changes have provided clues and constraints in discriminating between the causative faults of strong seismic events [24,25,39,40]. Therefore, in this paper, we focus on the hydrological changes originated by some historical earthquakes that have occurred in Central-Southern Italy (Figure 1) through: (i) the description of the methodologies of data retrieval; (ii) the classification of the data; (iii) an analysis of the potential applications and outcomes.

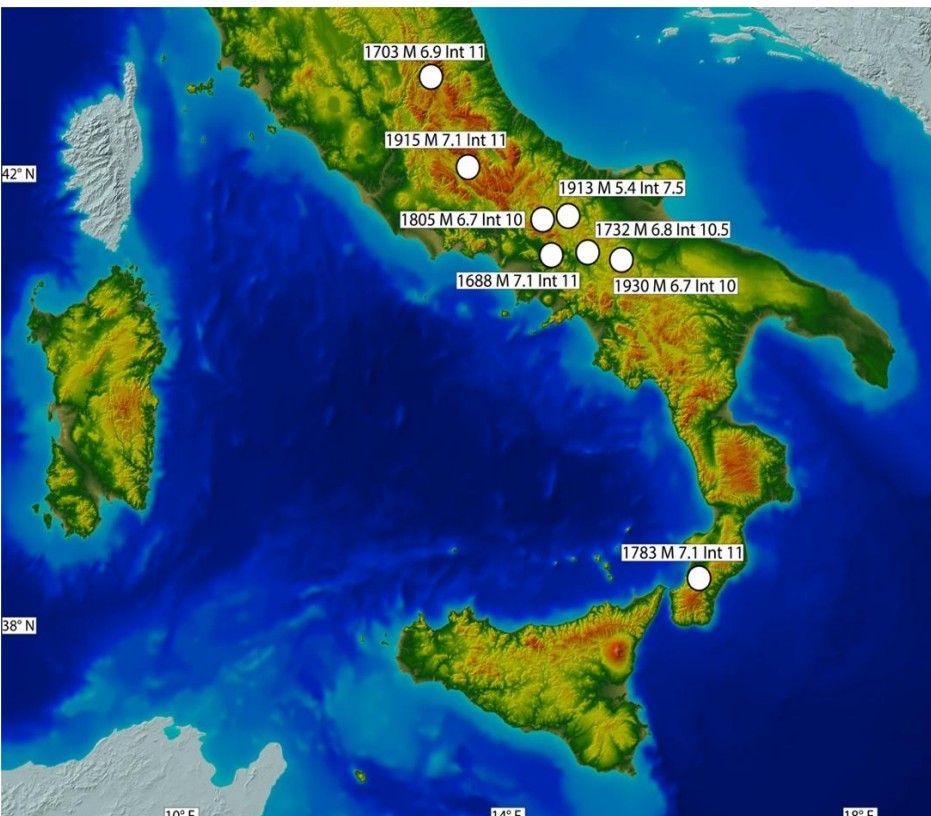

**Figure 1.** Location of the historical seismic events for which we show examples of data of earthquake-induced hydrological changes. Moment magnitude and MCS intensity are from the CPTI Italian Seismic Catalogue [41].

## 2. Materials and Methods

Similar to traditional research on historical seismology, the data retrieval of earthquake-induced hydrological effects becomes more difficult the further one goes back in time. Unlike the information used to estimate macroseismic intensity (mostly data on damage), observations on the hydrological effects are rarely subject to systematic collection (for very recent earthquakes, the situation is different; see the end of this section). Therefore, the research method differs depending on whether the object of the investigation is an ancient or a modern earthquake.

- 'ancient' earthquakes (~1600–~end of 18th Century)

Information on hydrological changes caused by earthquakes that occurred up until the second half of the 18th Century (Figure 2) is essentially qualitative and very difficult to retrieve. This kind of information is often 'hidden' in a variety of reports such as: chronicles, private letters and diaries, and local reports. For most of them, the event is mentioned as a minor curiosity and is eventually not even referred to individual earthquakes.

In Piedimonte, Terra del Signor Duca di Lorenzano, si fermò per buona pezza il corso di due grosse forgenti, che hanno lor'origine dalle Montagne di quel luogo, e formando unite un fiume servono a quei artegiani nella fabbrica de'panni, e carta. M'afferma il Signor Giacopo Fo-

In the village of Piedimonte, land of the Duke of Lorenzano, the flow stopped for a long while from two large springs that originate from the mountains around, and merging in a river are used by the artisans who work in the making of fabric, and of paper.

§. VIII. Ad orientem Lacus, in Territorio Patriæ meæ, in loco demaniali Univerſitatis ejuſdem, ubi diċitur *La Palombara*, & in latere orientali demanialis defenſæ; fons aquæ falſæ excurrit, cujus aquæ ſalem comunem, cum partibus aluminoſis decantatæ deponunt, diſtatque a Lacu, uno milliario, vel circa: & inter orientem, & Meridiem ad Eſt ſud Eſt Montes Terræ *Guardiæ Lombardorum* extenduntur, ad quorum radices ſcatent aquæ Torrentis, vulgo diċti *Fredene*, ut in Fig. V. qui poſt varios flexus immittitur in flumen, vulgo diċto *Calore*, quod intelligas, de *Calore* Hirpinorum, ne confundatur cum *Calore* Lucaniæ; & ex præfata ſcaturigine, in Anno 1732., poſt terræmotum, per plures dies aquas rubras continuo emiſſas concives mei teſtantur, ipſaſque ſulfure redolentes. Et ulterius, in Territorio *Melfitano*,

...and from that spring, in 1732, after the earthquake, my fellow citizens witness that for several days reddish water was flowing, and with sulphur smelling.

**Figure 2.** Examples of reports of hydrological observations following ancient earthquakes. Up: 1688 M = 7.1 Sannio event [42]. Down: 1732 M = 6.8 Irpinia event [43].

Infrequent sources of data derive from administrative correspondence between citizens and institutions for compensation claims, which are preserved in repositories and public archives (Figure 3).

Né solamente queste, Beatissimo Padre, sono le disgrazie che rende miserabile Cascia con il suo territorio poiché ha patito e patisce la siccità del fiume havendo l'acqua a cagione delle reiterate scosse de' terremoti preso altrove quasi totalmente il suo corso. Si riconoscono traviate le vene talmente che per molto tempo il letto del fiume è stato affatto asciutto. Da ciò, Padre Santo, si calcola un danno universale di [scudi] 3.000 essendo non solo rimasti inutili dieci molini da grano, ma di più di alcun frutto quasi lo spatio di otto miglia di prataria, quindi ne è risultato anco la

...and not only these, most blessed Father, are the misfortunes that make Cascia and its territory miserable as [the village] suffered and suffers the drought having the water almost completely changed its course because of the several earthquake shocks. Water veins are spoilt for the long time the river bed was dry. For this, Holy Father, a total loss of 3000 [escudos] is estimated as ten grain mills are now unused...

**Figure 3.** Excerpt from the memorial of the inhabitants of the city of Cascia to the Pope Clemente XI about the extension of the tax exemption granted after the 1703 earthquake in Central Italy [31].

Towards the end of the 18th Century, the first examples of proto-maps and sketches appear, reporting generic annotations of hydrological evidence, as a tread of the more detailed observations and images produced in the next few decades (Figure 4).

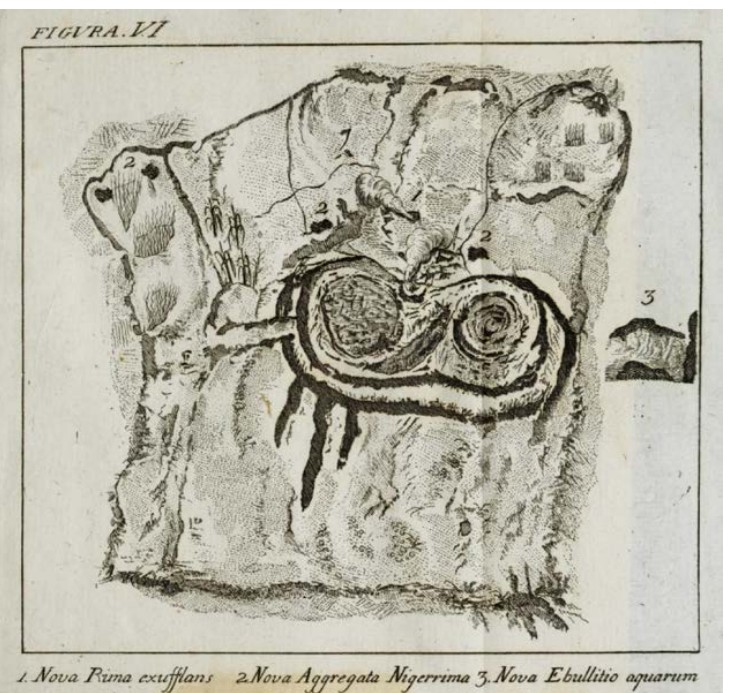

*1. Nova Prima exufflans   2. Nova Aggregata Nigerrima 3. Nova Ebullitio aquarum*

**Figure 4.** Sketch of the Lago delle Mefite d'Ansanto area carried out by V.M. Santoli [43], a priest, archeologist, and scholar who was the best witness of the time regarding that area. The analysis of Santoli's work in a geomorphological key has allowed, in modern times, a fruitful comparison with the current territorial arrangement, highlighting the permanence and evolution of the environment [44]. This area is the same as cited for the 1732 earthquake in Figure 2.

- 'modern' earthquakes (~end of 18th Century—1950)

For seismic events that have occurred in the past couple of centuries, research has usually benefitted from the growing interest in naturalistic observations by scholars, who, especially in the period from ~1800 to the first half of the 1900s, collected and classified effects observed in conjunction with earthquakes. These types of observations include—besides the damage—atmospheric phenomena, the behavior of animals, geomorphological modifications, and variations in hydrological regimes (streamflow discharge, springs, wells, liquefactions) e.g., [45,46].

This information, at times with no quantitative elements, is often found mixed and superimposed within generic studies on seismic events. Two large magnitude seismic events mark the divide of this renewed interest, the 1783 M = 7.1 Southern Calabria and the 1805 M = 6.7 Central Italy earthquakes. For the first time, several travel diaries, consistent between the coeval investigators, reported a great number of pre-, co-, and post-seismic hydrological observations clearly associated with the occurrence of the events [15,16,31,42,43,47–51] (Figure 5).

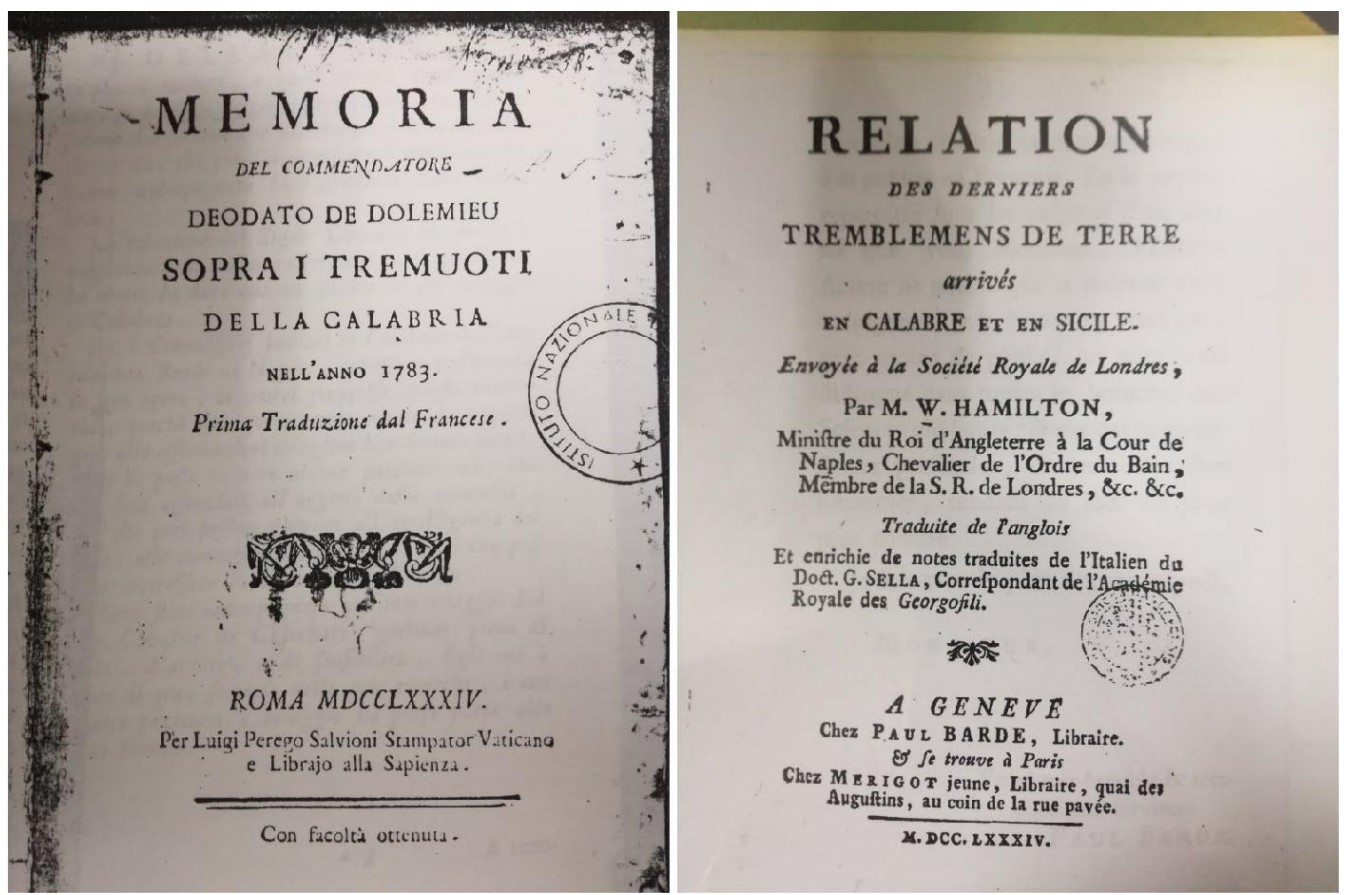

**Figure 5.** 1783 M = 7.1 Calabria earthquake. Reports by [47,48].

In particular, the detailed engravings and drawings produced by the members of the expedition sent to the 1783 epicentral area on behalf of the Kingdom of Naples (Figure 6) represent a faithful picture of the effects observed in the field, exploited in different studies still today [21,25].

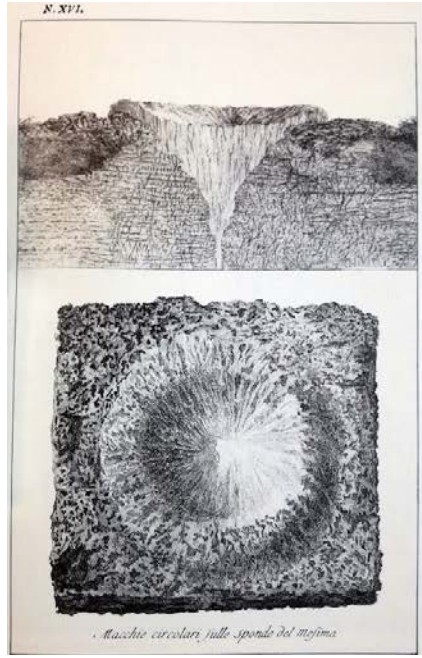 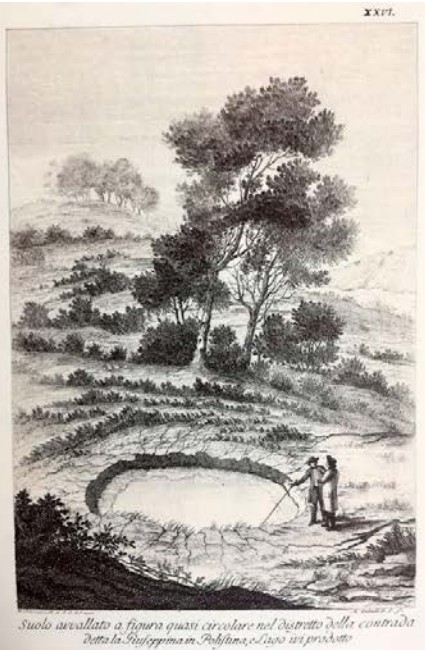 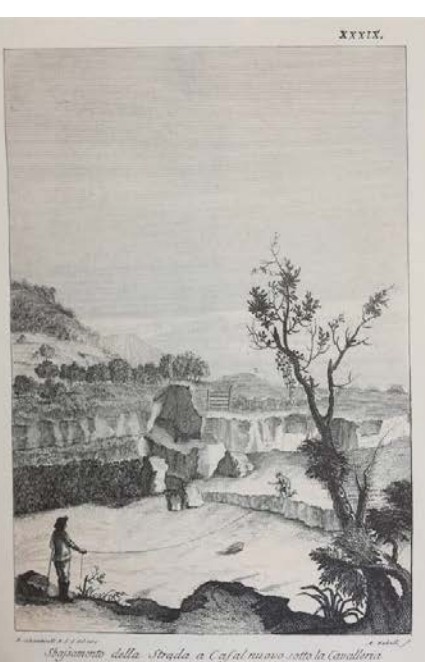

**Figure 6.** 1783 M = 7.1 Calabria earthquake. Reproduction of drawings carried out in the field after the earthquake by [49]. The original captions of the images reported 'circular spots on the banks of River Mesima' (**left**); 'quasi circular depression of the ground near Polistena, and lake there produced' (**center**); 'lowering of the road to Casalnuovo' (**right**). Modified after [25].

The reports following the 1805 earthquake, which struck the Central-Southern Apennines after a one-century-long period of relative seismic quiescence [41], confirm this 'new age' of scientific interest, with unprecedented detailed descriptions of hydrological effects (Figure 7) [16,50,51].

60. Nè furono meno ragguardevoli i fenomeni riguardanti le acque; perciocchè fin dal giorno precedente al Tremuoto le acque delle fontane di Bojano naturalmente fredde trovaronsi di avere acquistato un certo grado di tiepidezza, ed osservossi torbida la sorgente del fiume Trigni, che passa per la detta Città. In Isernia disseccaronsi le grandi sorgenti di acqua, che per via di un superbo canale costrutto dagli antichi Romani vi s'intromettono; e 'l gran rivo, che passa per Agnone, onde formasi poi il fiume Trigni, s'inaridì.

...and the phenomena regarding the waters were not less remarkable; because since the day before the earthquake the normally cold waters from the fountains in Bojano acquired a kind of tepidity, and turbid water was observed from the spring of River Trigni that flows in that town. In Isernia the large springs of water, that enter the town with a magnificent canal which was built by the Romans, turned dry; and the huge stream that flows in Agnone and forms the River Trigni dried up.

**Figure 7.** 1805 M = 6.7 Central Italy earthquake. Excerpt from the report by [16].

Starting from the second half of the 18th Century, an increasing set of data on earthquake-induced hydrological effects was provided by local and national newspapers, which started a frequent and widespread coverage of the news coming from the territory (Figure 8). The descriptions are more detailed and revealing of the nature of the effects.

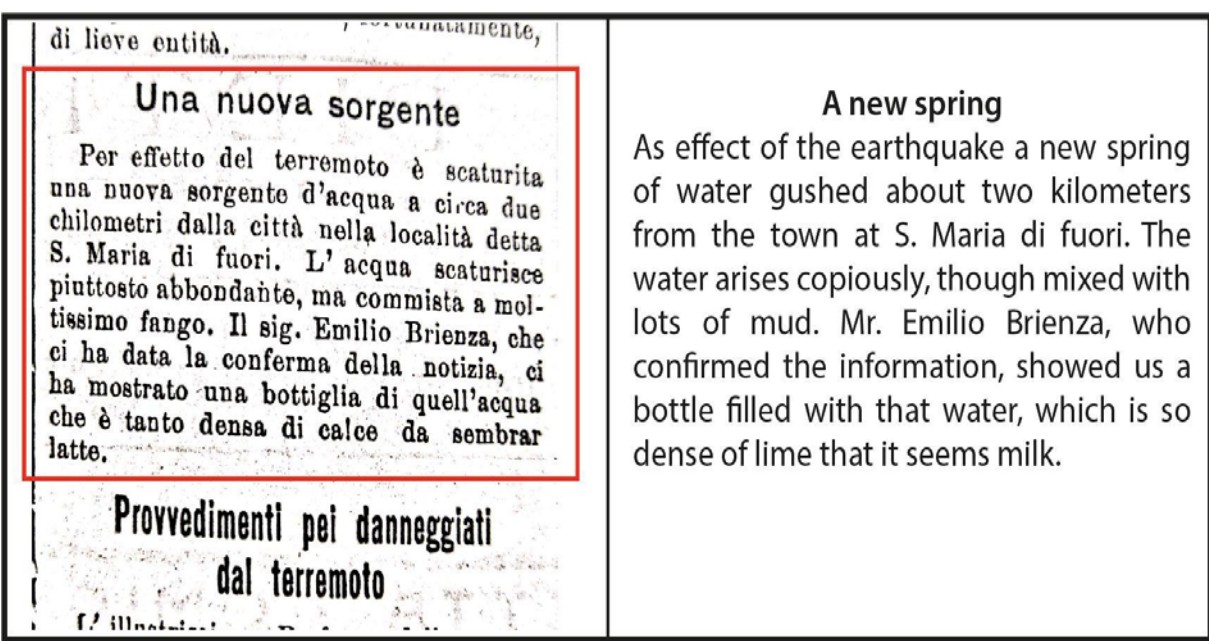

**Figure 8.** 1913 M = 5.4 Molise earthquake. Excerpt from the newspaper 'La Provincia di Campobasso'.

The last decades of the 18th Century saw the onset of new means of data retrieval, i.e., seismic postcards and, later, the macroseismic questionnaires, which provided thousands of data on earthquake effects that allowed a systematic and modern approach to earthquake studies. In particular, the seismic postcards (synthetic, standard reports of intensity from selected correspondents like municipalities, post offices, and police agencies that were collected until the end of the next century), explicitly ask the compiler for information about hydrological effects (Figure 9).

The last important branch of data retrieval comes from the aqueducts service and from the 'Annali Idrologici' (Hydrologic Annals), that provide regular and frequent observations and measures of the discharge of rivers and springs, of the level in wells, as well as a daily bulletin on rainfall. The Annali Idrologici have provided information since the 1920s, and are a sound and quantitative support to the collection of hydrological data associated with earthquakes (Figure 10), also in present times.

Aside from historical seismology, during the most recent decades, the observations of earthquake-induced hydrological effects have become a subject of systematic collection. Though, from a qualitative point of view, the information that is collected has not changed through the centuries, today, modern instruments and methods of recording provide data that is routinely used in different studies (e.g., [38,52–58]). As an example, an accurate survey of old and new datasets allows us to contribute to the study on the nature and role of fluids in the seismogenic processes and for hazard estimation, i.e., areas prone to liquefaction phenomena [59].

*Classification of Data*

Once the observations of earthquake-induced hydrological changes have been collected, there is the need to organize the data and to supply information and parameters to provide comprehensible access to scientists and other users, i.e., to envision the format and build up the final structure of folders and files for the storage of data and sources. The classification scheme we show in the following has been realized within the activities of the Project Further (The role of FlUids in the pReparaTory pHase of EaRthquakes in Southern Apennines) the aim of which is to explore, through multidisciplinary methods, the role of fluids in the different phases of the earthquake occurrence. Observational data coming from historical sources and from the literature are filed by earthquake, for which hydrological changes have been observed.

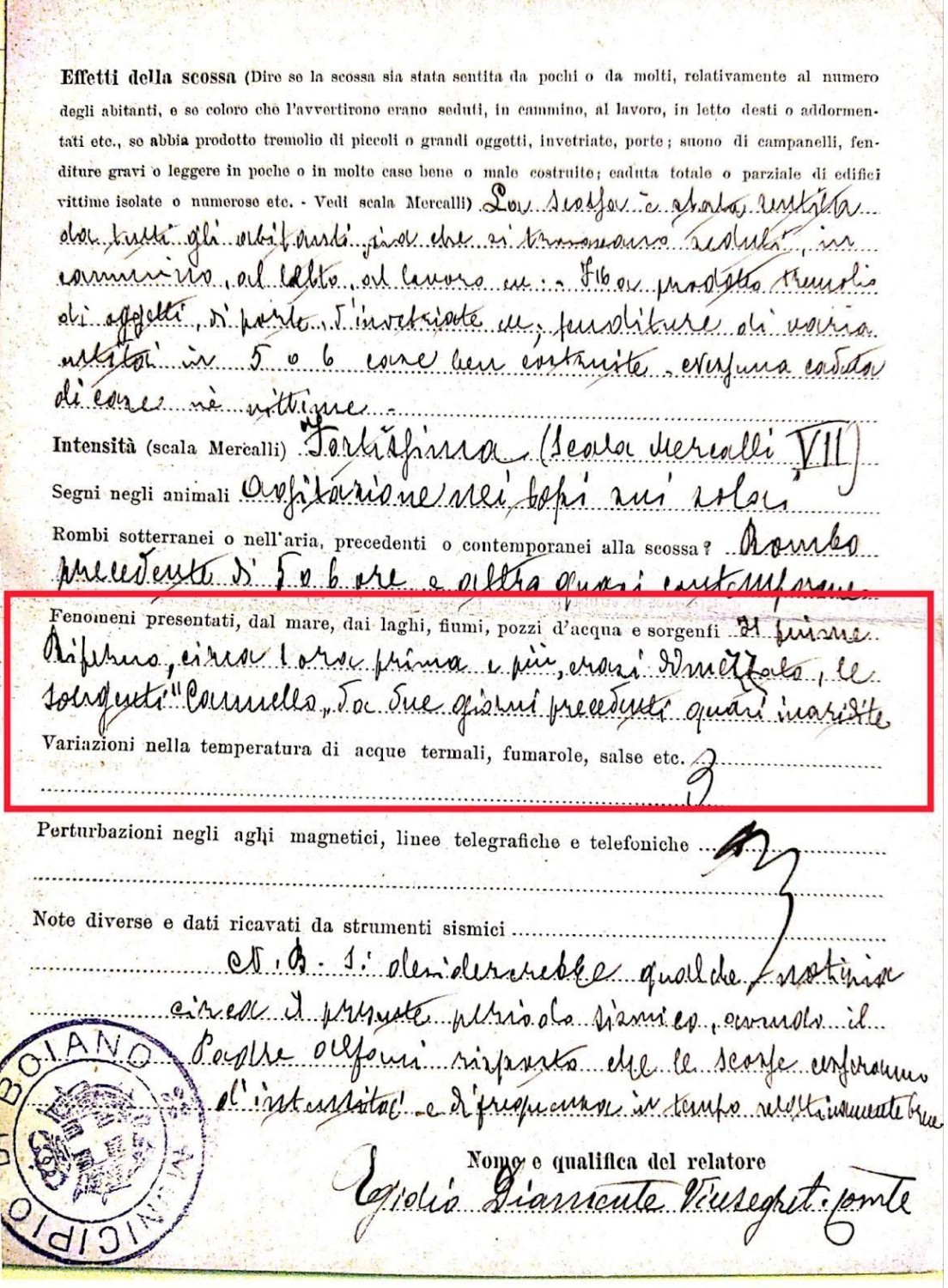

**Figure 9.** Example of seismic postcard from the 1913 M = 5.4 Molise earthquake. The information reported in the red square reads: 'The River Biferno, about one hour before [the earthquake] and more, had been halved, the Cannello springs, almost dried up two days earlier'.

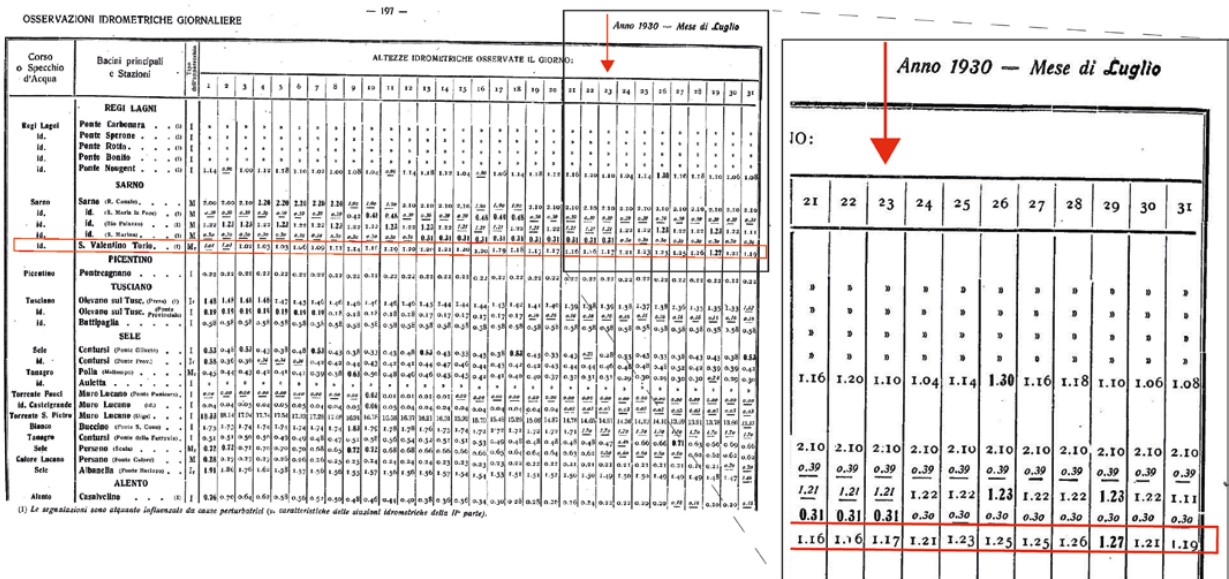

**Figure 10.** 1930 M = 6.7 Irpinia earthquake. Excerpt from the Annali Idrologici of 1930, showing a sharp spring discharge variation starting the day after the seismic event that occurred on 23rd July (indicated by a red arrow).

Within the abovementioned project, each study event is marked by a single folder, in which the fundamental product is the database of hydrological observations (Figure 11). Each observation is accurately reconsidered and localized, cross-checking the original description with topographic maps at different scales. In addition to the location and the geographical coordinates of the observation, we also report the epicentral distance from the seismic event and the most detailed description of the type of evidence of the hydrological effect, including some complementary notes by the compiler that provide further information about the reliability of the observation. The field 'References' in the database reports all of the sources that contributed to the fulfillment of the record. Based on the reliability of each reference/account, a three-fold quality code (A/B/C) is assigned to the data. The best rank 'A' is assigned to the observations that were clearly described and consistently reported by coeval investigators; second and third ranking is given to observations not clearly described or with tenuous evidence ('B'), or mislocated and/or doubtfully associated with the event ('C'). Needless to add, the oldest data are those with the lowest reliability, but, being the rarest, they are also the most valuable.

| NO. | Locality | LAT | LON | EP. DIST. (km) | EVIDENCE | CODE | NOTES | REFERENCE |
|---|---|---|---|---|---|---|---|---|
| 1 | Alife | 41.325 | 14.335 | 20 | chasm with emission of sulphur water | A | | Conforto 1895 |
| 2 | Alvignano | 41.275 | 14.347 | 18 | turbid flow from River Volturno | A | temporary phenomenon | Magnati 1688 |
| 3 | Alvignano | 41.275 | 14.347 | 18 | decreased discharge from River Volturno | B | temporary phenomenon - generic information | Magnati 1688 |
| 4 | Atella | 40.876 | 15.652 | 102 | gas and fumes exhalations | A | | Magnati 1688; Vera 1688 |
| 5 | Benevento | 41.126 | 14.775 | 25 | increased streamflow of River Sabato | C | doubtful attribution to the seismic event | ASRM 1691a; ASRM 1691b |
| 6 | Benevento | 41.146 | 14.830 | 27 | appearance of streamflow of bituminous water | A | Ponte Valentiniano (Apice in CFTI); lasted few days | Magnati 1688; Bulifon 1697 |
| 7 | Cerreto S. | 41.286 | 14.559 | 1 | chasm with emission of sulphur water in the mountains | A | | Conforto 1930; Mazzacane 1911 |
| 8 | Cerreto S. | 41.286 | 14.559 | 1 | bubbling soil in the mountains | B | | Mazzacane 1911 |
| 9 | Napoli | 40.848 | 14.261 | 55 | increased level in wells | B | generic information; preseismic observation | Pacichelli 1690 |
| 10 | Napoli | 40.848 | 14.261 | 55 | decreased flow from public fountains | B | generic information | Magnati 1688 |
| 11 | Napoli | 40.848 | 14.261 | 55 | increased flow from public fountains | B | generic information | Magnati 1688 |
| 12 | Napoli | 40.848 | 14.261 | 55 | variations in the chemical-physical characteristics of waters | B | generic information | Magnati 1688 |
| 13 | Napoli | 40.880 | 14.330 | 49 | appearance of streamflow of bituminous water | A | loc. Paludi "Acqua della Bufala" - subsequently dried | Conforto 1895; Bulifon 1697 |
| 14 | Piedimonte M. | 41.365 | 14.380 | 18 | during the shock the Torano river disappeared at the springs | A | | Vera 1688; Pacichelli 1690 |
| 15 | Piedimonte M. | 41.365 | 14.380 | 18 | after the shock increased streamflow of River Torano at the springs | A | | Vera 1688 |
| 16 | Piedimonte M. | 41.365 | 14.380 | 18 | gas exhalation and increased temperature of Torano springs | A | | Vera 1688; Pacichelli 1690 |
| 17 | Piedimonte M. | 41.356 | 14.367 | 18 | two important springs dried up | A | "the flow stopped for a long while from two large springs" | Bulifon 1697 |
| 18 | Pietraroia | 41.347 | 14.549 | 7 | gas exhalations | B | doubtful interpretation | Magnati 1688 |
| 19 | Vitulano | 41.165 | 14.660 | 16 | gas exhalations | A | loc. Valle di Vitulano | Bulifon 1697 |

**Figure 11.** Example of dataset of the hydrological effects associated with the 1688 M = 7.1 Sannio event. Citations: Conforto 1895 [60]; Magnati 1688 [61]; Vera 1688 [62]; ASRM 1691a [63]; ASRM 1691b [64]; Bulifon 1697 [42]; Conforto 1930 [65], Mazzacane 1911 [66]; Pacichelli 1690 [67].

In addition to the database of the observations, the folder associated with each event includes a sub-folder containing the original documents that were accessed and a text file for references, the bibliography, and transcriptions of all the information that was retrieved by the sources.

## 3. Results and Discussion

Databases of observations of earthquake-induced effects (both geomorphological and hydrological effects) are the first and most necessary outcome of the activity of data retrieval. Several databases exist of this type for the Italian territory, some all-embracing different types of observations [23,30,31], others dedicated to a single type of evidence (for instance, liquefaction: [28,32]). In addition to the database of the observations, the basic outcomes of this activity are maps of hydrological changes. As an example, we show the map of the earthquake-induced hydrological variations—obtained after research carried out within the Further Project—associated with the 1688 M = 7.1 Sannio seismic event (Figure 12; original dataset in Figure 11).

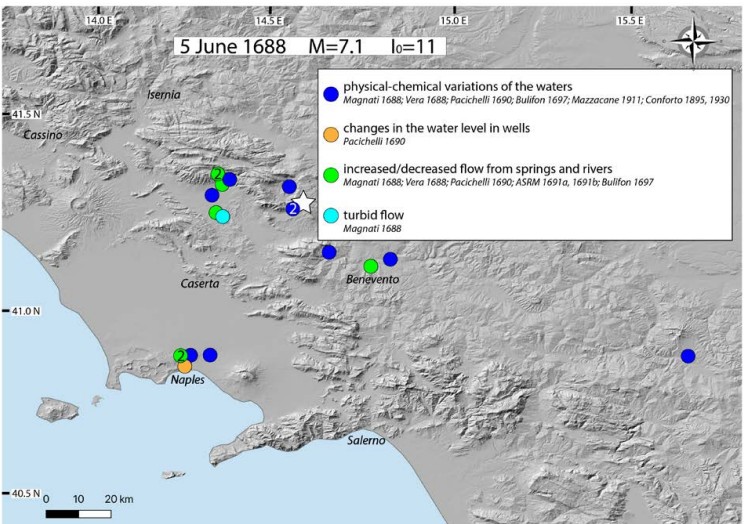

**Figure 12.** Map of the hydrological changes induced by the 1688 earthquake. Numbers inside the circles indicate multiple observations associated with the same location. A white star indicates the epicentral location from the CPTI Italian Seismic Catalog [41]. In the legend, we also indicate the sources from where the observations have been retrieved. Citations: Magnati 1688 [61]; Vera 1688 [62]; Pacichelli 1690 [67]; Bulifon 1697 [42]; Mazzacane 1911 [66]; Conforto 1895 [60]; Conforto 1930 [65], ASRM 1691a [63]; ASRM 1691b [64].

Considering that the most recent compilation of earthquake-induced effects in Italy [23,30,31] counts eleven observations relative to six locations for this event, it is worth noting that in-depth research allowed us to retrieve almost twenty data associated with nine different localities, which is a remarkable result given the very old age of occurrence of the earthquake.

Furthermore, in-depth research of earthquake-induced effects can provide scientific papers focused on individual, remarkable seismic events. These papers represent a fundamental input to update and implement databases. Table 1 shows a list of scientific papers that reported on earthquake-induced hydrological effects associated with the largest historical earthquakes that occurred in the Italian peninsula in the period 1650–1950.

**Table 1.** List of scientific papers that reported observations of hydrological effects induced by earthquakes that occurred on the Italian Peninsula in the period 1650–1950. Observations include variations in the discharge of springs and rivers, of the water level in wells, the formation and/or disappearance of springs, and changes in the chemical/physical characteristics of waters. Magnitude and intensity are from the CPTI Italian Seismic Catalog [41].

| Event | $M_w$ | $I_0$ | Reference | No. of Data |
|---|---|---|---|---|
| 24 July 1654 | 6.3 | 9–10 | Cucci and Cinti 2022 [68] | 4 |
| 5 June 1688 | 7.1 | 11 | Serva 1981 [69] | 6 |
| | | | This work | 19 |
| 8 September 1694 | 6.7 | 10 | Serva et al., 2007 [23] | 2 |
| 5 February 1783 | 7.1 | 11 | Porfido et al., 2011 [70] | 9 |
| | | | Cucci 2022 [25] | 25 |
| 26 July 1805 | 6.7 | 10 | Esposito et al., 1987 [71] | 45 |
| | | | Esposito et al., 2001 [72] | 30 |
| | | | Porfido et al., 2002 [73] | 48 |
| | | | Porfido et al., 2007 [22] | 29 |
| | | | Serva et al., 2007 [23] | 22 |
| | | | Cucci 2019 [24] | 77 |
| 2 January 1831 | 5.5 | 8 | Porfido et al., 1988 [74] | 1 |
| 9 April 1853 | 5.6 | 8 | Porfido et al., 1988 [74] | 1 |
| 16 December 1857 | 7.1 | 11 | Mallet 1862 [17] | 19 |
| 8 September 1905 | 6.9 | 10–11 | Tertulliani and Cucci 2009 [39] | 56 |
| | | | Porfido et al., 2011 [70] | 44 |
| 28 December 1908 | 7.1 | 11 | Comerci et al., 2015 [75] | 42 |
| | | | Guidoboni et al., 2018 [31] | 15 |
| 7 June 1910 | 5.8 | 8 | Martinelli 1913 [76] | 13 |
| 13 January 1915 | 7.1 | 11 | Cucci and Tertulliani 2015 [40] | 66 |
| 29 June 1919 | 6.4 | 10 | Guidoboni et al., 2018 [31] | 19 |
| 7 September 1920 | 6.5 | 10 | Guidoboni et al., 2018 [31] | 38 |
| 23 July 1930 | 6.7 | 10 | Esposito et al., 2001 [72] | 65 |
| | | | Porfido et al., 2007 [22] | 39 |
| | | | Serva et al., 2007 [23] | 37 |
| | | | Esposito et al., 2009 [20] | 39 |
| | | | Esposito and Porfido 2010 [77] | 40 |
| | | | Tranfaglia et al., 2011 [78] | 39 |
| | | | Cucci 2019 [24] | 46 |

Databases are usually the starting point for subsequent elaborations in hazard analyses and in seismotectonics. Among the former, knowledge of the history of occurrence of environmental effects after a seismic event helps in the identification of the areas where anthropic settlements and infrastructures are more exposed to this source of potential hazard. For instance, data coming from observations of liquefaction and/or differential shaking can supply an important input that integrates the traditional methods of seismic hazard assessment [79,80]. Liquefaction is a frequent damaging phenomenon, also during moderate magnitude earthquakes [29], so zonation products of liquefaction hazard are useful tools to identify areas that may display such soil instability [81,82].

Among the studies in seismotectonics, it is worth mentioning the studies that place the coseismic hydrological changes in relation to poroelastic theory [34], thus providing critical constraints of the style of faulting of major historical earthquakes that produced a large and

documented number of hydrological effects. Research in this direction led to discriminations among alternative causative faults of a number of strong historical earthquakes, which occurred in Central and Southern Italy [24,25,39,40,68]. Figure 13 shows an example of such elaborations relative to the 1915 M = 7.1 Fucino earthquake in Central Italy [40].

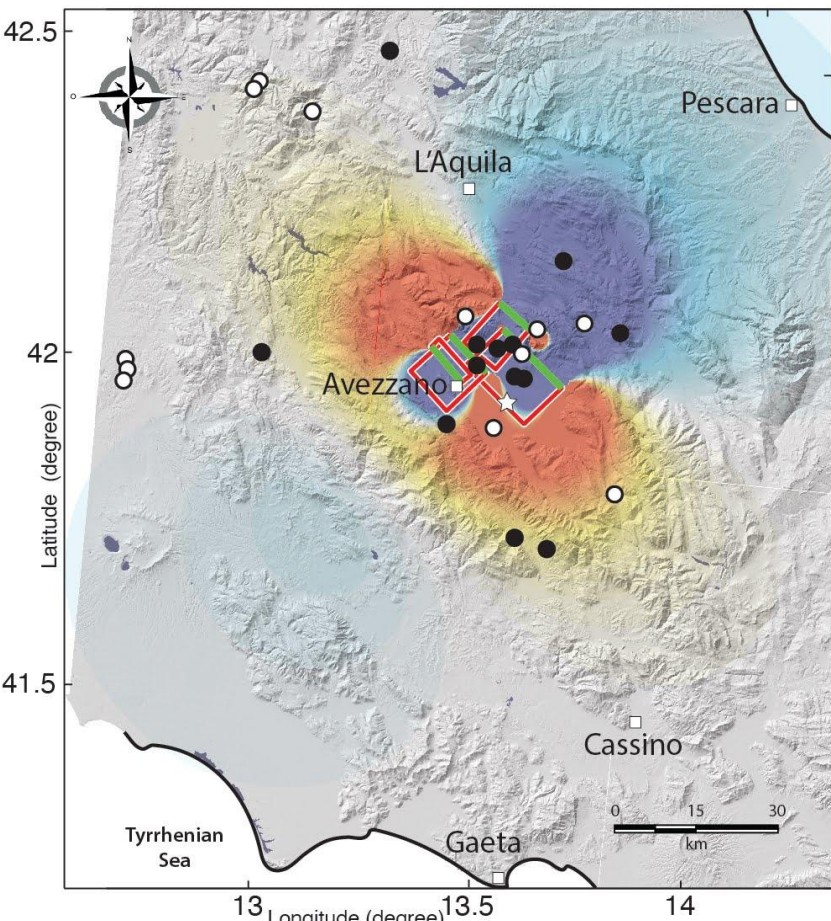

**Figure 13.** This map (modified after [40]) shows the location of the most likely seismogenic source of the 1915 M = 7.1 Fucino earthquake (epicenter indicated by a white star) on the basis of the pattern of the hydrological changes induced by the seismic event (black dots, increase in discharge; white dots, decrease in discharge). The hydrological observations derive from historical reports, seismic postcards, journals, and scientific papers. Blue shading indicates areas in compression, and red shading areas in dilatation. A red rectangle indicates the surface projection of the fault plane; a green line is the intersection of the updip projection of the fault with the surface. An increase in discharge is expected in compressional areas, and a streamflow decrease in dilatational areas. More details about the methodology of this research can be found in [24,40].

A recent outcome that has been developed during the last few decades is to consider the hydrological effects for assessing macroseismic intensity according to the guidelines of the ESI 2007 scale [14,23], which aims to integrate the intensity evaluations based on the traditional macroseismic scales.

The research of information associated with earthquakes that occurred several centuries ago can be frustrating and time consuming. This is particularly true when the subject of the research does not concern the most common information regarding the damage inflicted, but rare and sporadic observations of hydrological variations. However, the favorable combination of an important seismic history with a detailed written history that is available in several archives and repositories allows unexpected results, even for very old or moderate magnitude earthquakes. In this paper, we presented a comprehensive

view of the different types of sources where investigators can find such information, as well as a scheme for the classification of data. A first and the most common use for this data satisfies the need and the natural tendency to increase, develop, and classify our knowledge in the form of specific catalogs, maps, and scientific papers related to individual seismic events. However, we maintain that the parametrization of the hydrological changes induced by historical earthquakes can provide hints, even to seismotectonic analyses and to local hazard estimates.

**Author Contributions:** C.C., L.C. and A.T. have equally contributed to the paper. All authors have read and agreed to the published version of the manuscript.

**Funding:** This research was funded by the Earthquake Department Strategic Project FURTHER (The role of FlUids in the pReparaTory pHase of EaRthquakes in Southern Apennines) at the Istituto Nazionale di Geofisica e Vulcanologia (INGV).

**Data Availability Statement:** Not applicable.

**Acknowledgments:** Thanks to three reviewers who provided important comments and suggestions to the manuscript. Thanks to the Academic Editor, the Handling Editor, and to the Assistant Editor. The methodology described in the present paper is adopted within the FURTHER Project (The role of FlUids in the pReparaTory pHase of EaRthquakes in Southern Apennines), by the Istituto Nazionale di Geofisica e Vulcanologia (INGV), with the goal of establishing the role of fluids in the preparatory phase of earthquakes in the Southern Apennines. In particular, the Working Package 'Historical seismicity' deals with the retrieval of hydrological data associated with earthquakes that occurred since ~1600 in three key areas of the Southern Apennines.

**Conflicts of Interest:** The authors declare no conflict of interest.

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
