# Peer review of "Reappraisal of Data of Hydrological Changes Associated with Some Strong Historical Italian Earthquakes"

_geosciences, doi:10.3390/geosciences13020055_

Round 1
Reviewer 1 Report
I consider the work valid, and worthy of publication as it summarizes a useful methodology adopted by the authors over the years to study historical earthquakes.
Some points should be better dealt with, methodologies better developed, and perhaps the work could be made more useful to readers by adding data as supplementary material.
The work can be considered for publication with moderate revisions.
See attached pdf file for comments.

Reviewer 2 Report
This aim of this paper is studying the hydrological changes, occurred by historical earthquakes, based on the description of the methodologies of data retrieval, the data classification and the analysis of the potential applications and outcomes.
The quality of the paper is not satisfactory, while necessary sections have not been included (“Results” and “Discussion” sections are missing). Moreover, significant omissions are observed. In particular:
Lines 53: Before “Materials and Methods” section, I suggest adding a new section, named “Geodynamic setting”. In this section, the active tectonic setting should be described, which is directly associated with earthquake occurrences.
Line 146: The “Classification of data” section should be merged with the “Materials and Methods” section.
Line 170: I suggest renaming the “Outcomes and applications” section into “Results”, while a map of the major historical earthquakes should be included, before Figure 11.
Line 176: Explain why the 1688 M=7.1 Sannio seismic event was preferred as an example. More details should be provided.
Line 208: The “Conclusions” section resembles an abstract rather than conclusions. It should be rewritten, while the major findings should be highlighted.
General comment: Significant omissions are observed in References (e.g. Ambraseys, 2009), while the “Discussion” section is missing.
Unfortunately, the entire paper should be rewritten and therefore it is rejected.
Round 2
Reviewer 2 Report
The initial review is resubmitted below, checking the implementation of the proposed corrections:
The aim of this paper is studying the hydrological changes, occurred by historical earthquakes, based on the description of the methodologies of data retrieval, the data classification and the analysis of the potential applications and outcomes.
The quality of the paper is not satisfactory, while necessary sections have not been included (“Results” and “Discussion” sections are missing). Moreover, significant omissions are observed. In particular:
Lines 53: Before “Materials and Methods” section, I suggest adding a new section, named “Geodynamic setting”. In this section, the active tectonic setting should be described, which is directly associated with earthquake occurrences. Please, modify. It has not been implemented in the new version.
Line 146: The “Classification of data” section should be merged with the “Materials and Methods” section. Please, apply. It has not been implemented in the new version.
Line 167: This a Table and it should not be presented, as a Figure. Please, modify it, based on the template, provided by the journal. It has not been implemented in the new version.
Line 170: I suggest renaming the “Outcomes and applications” section into “Results”, while a map of the major historical earthquakes should be included, before Figure 11. Please, apply. It has not been implemented in the new version.
Line 176: Please, explain why the 1688 M=7.1 Sannio seismic event was preferred as an example. More details should be provided. Please, apply. It has not been implemented in the new version.
Line 179: The resolution of Figure 11 should be improved. Please, apply. It has been implemented in the new version.
Line 180: Please, determine in Figure 11 caption the star symbol and number “2”, observed within some circles. . It has been partially implemented in the new version.
Line 208: The “Conclusions” section resembles an abstract rather than conclusions. It should be rewritten, while the major findings should be highlighted. It has not been implemented in the new version.
General comment: Significant omissions are observed in References (e.g. Ambraseys, 2009) It has not been included in the new version, while the “Discussion” section is missing. The aforementioned comments should be considered and applied by the authors. Unfortunately, the entire paper should be rewritten.
As mentioned above, the majority of the proposed corrections has not been considered. However, the structural problem is that the paper has been characterized as a “Review paper” by the authors, which is not. Initially, the paper structure does not meet the criteria of a “Review paper”; it follows the “Research article” structure, while the significant omission of references remains (Review papers include numerous references, more than 150; in this paper 60 references have been considered). Finally, my proposal was to rewrite the paper in order to receive an acceptable form. Unfortunately, I have to reject the paper again, based on all the above.
